# Investigation of Different Methods of Intraoperative Graft Perfusion Assessment during Kidney Transplantation for the Prediction of Delayed Graft Function: A Prospective Pilot Trial

**DOI:** 10.3390/jpm12101749

**Published:** 2022-10-21

**Authors:** Andreas L. H. Gerken, Michael Keese, Christel Weiss, Hanna-Sophie Krücken, Katarina A. P. Pecher, Augusto Ministro, Nuh N. Rahbari, Christoph Reissfelder, Ulrich Rother, Babak Yazdani, Anna-Isabelle Kälsch, Bernhard K. Krämer, Kay Schwenke

**Affiliations:** 1Department of Surgery, University Medical Center Mannheim, Medical Faculty Mannheim, Heidelberg University, Theodor-Kutzer-Ufer 1-3, D-68167 Mannheim, Germany; 2European Center for Angioscience, Medical Faculty Mannheim, Heidelberg University, Ludolf-Krehl-Straße 13-17, D-68167 Mannheim, Germany; 3Department of Biometry and Statistics, University Medical Center Mannheim, Medical Faculty Mannheim, Heidelberg University, Theodor-Kutzer-Ufer 1-3, D-68167 Mannheim, Germany; 4Lisbon Academic Medical Centre, 1649-035 Lisbon, Portugal; 5Vascular Surgery, Heart and Vessels Department, Hospital Santa Maria (CHULN), 1649-035 Lisbon, Portugal; 6Faculty of Medicine, University of Lisbon, 1300-477 Lisbon, Portugal; 7Department of Vascular Surgery, Friedrich Alexander University Erlangen-Nuremberg, Krankenhausstraße 12, D-91054 Erlangen, Germany; 8Department of Medicine V, University Medical Center Mannheim, Medical Faculty Mannheim, Heidelberg University, Theodor-Kutzer-Ufer 1-3, D-68167 Mannheim, Germany; 9Center for Innate Immunoscience, Medical Faculty Mannheim, Heidelberg University, Ludolf-Krehl-Straße 13-17, D-68167 Mannheim, Germany

**Keywords:** renal allograft, transplant function, near-infrared fluorescence, indocyanine green, laser Doppler, tissue oxygenation

## Abstract

Delayed graft function (DGF) after renal transplantation is a relevant clinical problem affecting long-term organ function. The early detection of patients at risk is crucial for postoperative monitoring and treatment algorithms. In this prospective cohort study, allograft perfusion was evaluated intraoperatively in 26 kidney recipients by visual and formal perfusion assessment, duplex sonography, and quantitative microperfusion assessment using O2C spectrometry and ICG fluorescence angiography. The O2C tissue spectrometry device provides a quantitative method of microperfusion assessment that can be employed during kidney transplantation as an easy-to-use and highly sensitive alternative to ICG fluorescence angiography. Intraoperative microvascular flow and velocity in the allograft cortex after reperfusion predicted DGF with a sensitivity of 100% and a specificity of 82%. Threshold values of 57 A.U. for microvascular flow and 13 A.U. for microvascular velocity were identified by an ROC analysis. This study, therefore, confirmed that impairment of microperfusion of the allograft cortex directly after reperfusion was a key indicator for the occurrence of DGF after kidney transplantation. Our results support the combined use of intraoperative duplex sonography, for macrovascular quality control, and quantitative microperfusion assessment, such as O2C spectrometry, for individual risk stratification to guide subsequent postoperative management.

## 1. Introduction

Delayed graft function (DGF) after kidney transplantation is an increasing clinical problem. The rising incidence rate of DGF correlates with the introduction of expanded criteria donors (ECDs) and donation after cardiac death (DCD) in the context of organ shortage [1]. DGF is known to occur in 25–35% of transplanted organs, but rates up to 50% have been reported [2,3,4,5]. ECD programs in Europe include the Eurotransplant Senior Program (ESP), the Recipient-oriented Extended Allocation Program (REAL), and the Rescue Allocation (RA). ECD kidneys might possibly be related to marginal organ quality [6] with a higher rate of postoperative graft dysfunction [7,8,9,10]. However, ECD kidney recipients still show better overall survival than patients remaining on dialysis therapy [9]. Since DGF is associated with impaired long-term graft survival, strategies to detect and reduce DGF incidence should be implemented to improve graft survival [2,11,12,13].

To date, the only intraoperative quality control officially requested by procurement authorities in Europe is visual assessment of the color after reperfusion and the amount of intraoperative urine production. Doppler ultrasound is commonly added to judge macrovascular perfusion of the renal allograft [14,15,16]. Different quantitative tools have been employed to assess graft perfusion after revascularization noninvasively in real time. Quantitative indocyanine green (ICG) near-infrared fluorescence angiography has been established for predicting DGF using the Spy Elite system (Stryker, Kalamazoo, MI, USA) [17,18,19]. Threshold values were defined for ICG Ingress, representing the quality of inflow. Intraoperative organ spectrometry with oxygen to see (O2C) (LEA Medizintechnik, Giessen, Germany) has been used to assess microperfusion mainly in limbs [20,21,22,23,24,25], but also in solid organs [26,27,28] and in kidney allografts [29,30].

The aim of this study was to evaluate different methods of intraoperative perfusion assessment for predicting DGF after kidney transplantation.

## 2. Materials and Methods

### 2.1. Inclusion Criteria and Patient Selection

All patients scheduled for kidney transplantation in the University Medical Center Mannheim (Germany) during a time period of 2 years (October 2019–September 2021) were screened for study inclusion (*n* = 54). Provided the absence of exclusion criteria, namely known allergy to iodine or ICG, severe hepatic dysfunction, pregnancy, hyperthyroidism, pulmonary hypertension, and the logistical availability of both technical tools for quantitative perfusion assessment (O2C and SPY Elite) and the existence of written informed consent, 26 patients were prospectively enrolled in the study (Figure 1). The study was conducted in accordance with the Declaration of Helsinki, was approved by the ethics committee of the Medical Faculty Mannheim, University of Heidelberg (2019-668N), and was registered (registration number DRK S00030097, DRKS—German Clinical Trials Register).

### 2.2. Study Design and Procedure

In this prospective pilot trial, standard techniques were applied for preoperative patient care, organ allocation via EUROTRANSPLANT (Leiden, The Netherlands), organ procurement, and the transplantation procedure. Postoperative care was conducted along a standardized treatment pathway [31] in all patients. After completion of the vascular anastomoses, graft perfusion was assessed visually via duplex sonography, O2C spectrometry, and ICG perfusion measurement by the transplant team. In parallel, cardiocirculatory parameters at the time of perfusion assessment were recorded.

### 2.3. Visual and Formal Perfusion Assessment

Formal evaluation of the “color after reperfusion” was performed as required by the German Organ Procurement Organization (DSO) and marked on the “Kidney Quality Form” as “homogeneous”, “marbled”, or “dark blue”.

### 2.4. Macroperfusion Assessment by Ultrasound

A sterile ultrasound probe (T-Shaped Intraoperative Transducer, I14C5T (9016), BK Medical ApS, Herlev, Denmark) was connected to the bk5000 Ultrasound System (CE0543, BK Medical ApS, Herlev, Denmark). The probe was placed directly on the surface of the graft. All anastomosis, the inflow and outflow vessels, as well as the cortex and parenchyma of the kidney were closely examined for stenosis, dissection, kinking of vessels, or lack of perfusion within all areas of the graft. The result was recorded by two experienced transplant surgeons.

### 2.5. Intraoperative Spectrometry of Allograft Microperfusion with Oxygen to See (O2C)

Cortical graft microperfusion was assessed quantitatively using the O2C device (Oxygen to see, version III, LEA Medizintechnik, Giessen, Germany). The device has been approved as a medical device class IIa according to regulation (EU) 2017/745. Micro-Lightguide O2C combines white light tissue spectrometry (wavelength range: 500–630 nm; optical resolution: 1.5 nm) and laser Doppler flowmetry (830 nm). The probe simultaneously measures the following four parameters of microcirculation from the tissue surface: postcapillary oxygen saturation of hemoglobin (SO2, in %), relative microvascular amount of hemoglobin (rHb, in A.U. (Arbitrary Units)), relative microvascular blood flow (flow, in A.U.) and microvascular blood flow velocity (velocity, in A.U.).

The intraoperative measurements were conducted with a sterile LFX-29 probe (catchment volume 3 mm) that was placed directly onto the surface of the allograft. The measurements were performed, according to a standardized protocol, 5 min after reperfusion at three different sites (upper, middle, and lower parts of the graft), each for a duration of 10 s. The parameter values were immediately shown on the display of the O2C device. The measurement results from the three sites were averaged to obtain a single value to represent the kidney. The measurements were only performed once, i.e., no serial scans were performed.

### 2.6. Intraoperative Fluorescence Angiography with ICG

Subsequently, the SPY Elite System (Stryker, Kalamazoo, MI, USA) was used for intraoperative fluorescence angiography to visualize cortical microperfusion of the graft, as described in previous studies [17,18,19]. The fluorescent dye ICG (Verdye, Diagnostic Green, Belgium) was injected in a standardized dose of 0.02 mg ICG per kg body weight. Quantitative assessment was performed with the integrated software (SPY-Q, Stryker). Four parameters are defined in the quantitative analysis of the intraoperative fluorescence videos: “Ingress” represents the difference between the baseline fluorescence intensity and the maximum intensity assessed; “Egress” is the difference between maximum intensity and final intensity; “IngressRate” quantifies the blood inflow by evaluating the increase in the fluorescence intensity per second (increase in gray stats per second); and “EgressRate” characterizes the outflow of blood, measured as the decrease in fluorescence intensity per second.

### 2.7. Clinical Parameters of Graft Function

The need for postoperative hemodialysis and the parameters of early kidney function (serum creatinine levels on postoperative days (PODs) 1–10; estimated glomerular filtration rate (eGFR) on PODs 1 and 7; cumulative diuresis in the first 24 h after transplantation; and diuresis on PODs 1, 2 and 7) were monitored. DGF was defined as the need for two or more sessions of hemodialysis postoperatively, according to Schnuelle et al. [32].

### 2.8. Statistical Analysis

All statistical calculations were performed using SAS statistical software, release 9.4 (SAS Institute Inc., Cary, NC, USA). Continuous variables are presented as mean values together with standard deviations or (e.g., in the case of skewed variables) as median values together with minimum and maximum. For categorical variables, absolute and relative frequencies (percentages) are given. The comparison of two independent groups (e.g., DGF versus non-DGF) was performed using Fisher’s exact test or the Mann–Whitney U test, as appropriate. The CKD-EPI equation was used to estimate GFR.

In order to investigate the correlation between two continuous variables of perfusion assessment, Pearson’s correlation coefficient was assessed.

Furthermore, logistic regression analyses were performed in order to identify parameters potentially associated with DGF. A receiver operating characteristic (ROC) curve was generated for the O2C parameters “flow” and “velocity”. For each ROC curve, the AUC (area under the curve) was estimated together with 95% confidence intervals.

For all statistical tests, *p* < 0.05 was considered to show a statistically significant test result.

## 3. Results

### 3.1. Patients and Procedure Characteristics

This prospective cohort study contains the assessment and associated data of 26 transplanted kidneys (22 deceased-donor kidney transplantations from donation after brain death and 4 living-donor kidney transplantations). Patient and donor characteristics are displayed in Table 1. At the time of perfusion assessment, the median heart rate was 65 (53–100) bpm, the median systolic blood pressure was 120 (100–136) mmHg, and the median rate of norepinephrine was 0.3 (0–1.0) mg/h.

### 3.2. Postoperative Results and Delayed Graft Function

DGF occurred in five patients (19%). None of the four patients after living-donor kidney transplantation developed DGF. None of the common perioperative parameters and comorbidities were found to be significantly associated with DGF, nor was the immediate urine production after reperfusion (none vs. moderate vs. good), or the type of donation or vascularization (Table 2).

The average postoperative creatinine levels of both groups were significantly different on PODs 2–10 (Figure 2a). There was no significant difference on PODs 1 (6.04 ± 1.89 mg/dL in DGF vs. 5.82 ± 1.77 mg/dL in non-DGF, *p* = 0. 8392). The average postoperative diuresis on days 0, 1, and 2 was significantly different (Figure 2b). There was no significant difference on POD 7 (median values 1750 mL in DGF vs. 2195 mL in non-DGF, *p* = 1.0000).

### 3.3. Formal and Visual Perfusion Assessment

Concerning the visual aspect of the graft after reperfusion (Table 3), 24 grafts were classified as “homogeneous” after reperfusion and two grafts were classified as “marbled”. There was no significant association between the intraoperative clinical parameters or the visual assessment of the color of the graft after reperfusion and DGF (OR = 5.00, 95%-CI: 0.26–97.70, *p* = 0.3538).

### 3.4. Perfusion Assessment by Ultrasound

In all grafts, perfusion was classified as “homogenous”. No major arterial, venous, or anastomotic problems were documented.

### 3.5. Association between Intraoperative Perfusion Analysis with ICG and Delayed Graft Function

In the DGF group, surprisingly, the average value for the absolute ICG perfusion parameter Ingress was higher, but the IngressRate characterizing the quality of inflow of blood into the kidney allograft was nearly half as low as in the non-DGF group. The outflow parameters Egress and EgressRate were lower in the DGF group. However, all of the four intraoperative ICG perfusion parameters were not associated significantly to DGF (Table 3).

### 3.6. Correlation of Intraoperative Perfusion Assessment with O2C and ICG

The following O2C parameters correlated significantly positively with ICG IngressRate: SO2 (r = 0.60275, *p* = 0.0497), flow (r = 0.69319, *p* = 0.0124), and velocity (r = 0.76678, *p* = 0.0036) (Figure 3). There was no significant correlation of the O2C parameter rHb with Egress or with EgressRate (*p* = 0.9180 and *p* = 0.2187, respectively).

### 3.7. Association between Intraoperative Perfusion Analysis with O2C and Delayed Graft Function

The mean values for the O2C parameters “flow” (Figure 4c) and “velocity” (Figure 4d) differed significantly between the DGF and non-DGF groups. For the parameters SO2 (Figure 4a) and rHb (Figure 4b), no significant difference between the groups was documented. The tests in Table 3 should be regarded as exploratory statistic. Because of the small sample sizes, we decided not to correct for multiple comparisons in order to obtain an impression of which parameters differ between the two groups and in order to avoid a rather large type 2 error.

The ROC analysis of the O2C perfusion parameter “flow” yielded an optimal cutoff value of 57.34 A.U., with a sensitivity of 100% and a specificity of 82% (AUC = 0.855, CI: [0.651; 1.000], *p* = 0.1299) for the prediction of DGF.

The ROC analysis of the O2C perfusion parameter “velocity” yielded an optimal cutoff of 13.33 A.U., with a sensitivity of 100% and a specificity of 82% (AUC = 0.891, CI [0.709; 1.000], *p* = 0.1421) for the prediction of DGF. However, these associations failed to reach statistical significance.

## 4. Discussion

The results of this prospective comparative cohort study suggest that even experienced transplant surgeons cannot predict DGF by means of intraoperative visual inspection or by duplex sonography. Quantitative perfusion assessment of microperfusion should be employed to enable an accurate individual risk stratification concerning the occurrence of DGF after kidney transplantation.

Concordant with the literature, the incidence of DGF in our study was about 19%. In our study population, the common risk factors were not associated with DGF. Despite the fact that postoperative serum creatinine levels and urine outputs were significantly different in patients with a normal postoperative graft function as opposed to DGF, they vary considerably in the early postoperative period. Therefore, these parameters may not allow for the prediction of short- and long-term graft function, or the individual risk stratification of DGF. Consequently, extensive postoperative monitoring is necessary for all kidney recipients until patients at risk can be identified.

As known from gastrointestinal procedures, the surgeons’ clinical judgement is of limited value in predicting postoperative outcome [33], presumably due to an overestimation of results. The use of duplex sonography, including more objective parameters, for estimating postoperative transplant function, such as the resistance index (RI), is being discussed controversially in the literature [18,34,35,36,37]. This is in line with our findings.

We assume that quantitative intraoperative assessment of the graft’s microperfusion after reperfusion is a more precise tool for predicting postoperative early graft function because it can be interpreted as a surrogate parameter reflecting the combination of risk factors present in an individual patient [19], organ quality [18], and the extent of the manifestation of ischemia-reperfusion injury (IRI). Impairment of the microperfusion of the allograft cortex seems to be a key risk factor for the occurrence of DGF [38,39,40].

Fluorescence angiography with ICG is an imaging modality with a predictive ability for DGF, as supported by prospective trials [18,19]. The Spy Elite System has been successfully employed for this purpose. A threshold value of 126.23 A.U. for the perfusion parameter ICG Ingress as an absolute parameter characterizing the peak inflow fluorescence intensity has been shown to predict DGF with a sensitivity of 78.3% and a specificity of 80.8% in 128 transplant patients [19]. However, not all systems allow for an additional quantification of ICG inflow and outflow, and heterogeneous perfusion parameters gained by different systems still need standardization [41]. Furthermore, this technique requires a comparable dosing of ICG and a certain level of training within the team [42]. Allograft spectrometry and laser Doppler microvascular blood flowmetry with the O2C device has emerged as a noninvasive alternative method for intraoperative microperfusion assessment [29,30]. It is comparably safe, is more cost-efficient, does not require potentially toxic or expensive contrast agents, and has a simpler logistical setup than that of fluorescence angiography. Specific threshold values for these O2C perfusion parameters were either not reported in previous studies or were calculated on the basis of measurements, with a different probe leading to a higher level of flow and velocity than in our results.

To the best of our knowledge, so far, there has not been a study comparing different modalities of intraoperative perfusion assessment during kidney transplantation with regard to a patient-specific prediction of DGF. In the present study, the ICG IngressRate correlated significantly with the O2C parameters flow and velocity. Consequently, these parameters had significantly lower values in the DGF group. We were able to deliver threshold values for the parameters flow and velocity, allowing for the prediction of DGF. The results of intraoperative risk stratification via O2C can be used to guide postoperative care. Patients with a flow exceeding 57 A.U. or a velocity exceeding 13 A.U. can be transferred to normal wards earlier, conserving valuable IMC/ICU space without endangering patient safety. On the other hand, kidney recipients with a critical microvascular flow or velocity should be monitored more closely, potentially on IMC units with a focus on systemic hemodynamics, renal blood flow, sufficient hydration, and early discussion of a percutaneous biopsy to rule out acute rejection, since IRI can trigger immunologic responses predisposing to acute graft rejection [43,44].

The O2C method is limited by the selective measurements and the depths of penetration. This seems to apply in particular to the parameters SO2 and rHb, derived from the tissue surface by white light spectroscopy. In case of a subcapsular hematoma or an abundance of fatty tissue, the measurements of superficial parameters might fail. Therefore, we still routinely combine microperfusion measurement with duplex sonography. Even in this comparably small cohort, the O2C method showed promising results.

## 5. Conclusions

This pilot trial delivers a solid basis for larger prospective trials with the aim of validating O2C threshold values and investigating the optimal point in time for employing intraoperative quantitative microperfusion assessment of the renal allograft for individual risk stratification concerning postoperative transplant function in order to guide postoperative management after kidney transplantation. O2C probe specifications should be reported exactly in further publications in order to allow for a comparison between different studies.

## Figures and Tables

**Figure 1 jpm-12-01749-f001:**
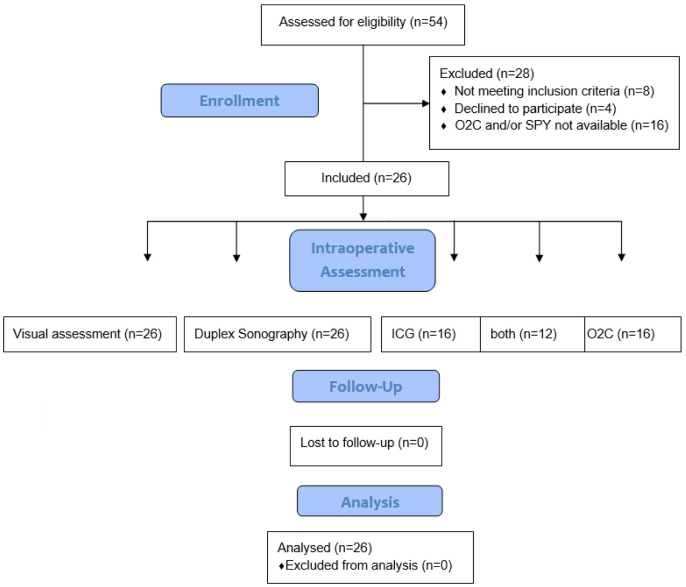
Flow diagram.

**Figure 2 jpm-12-01749-f002:**
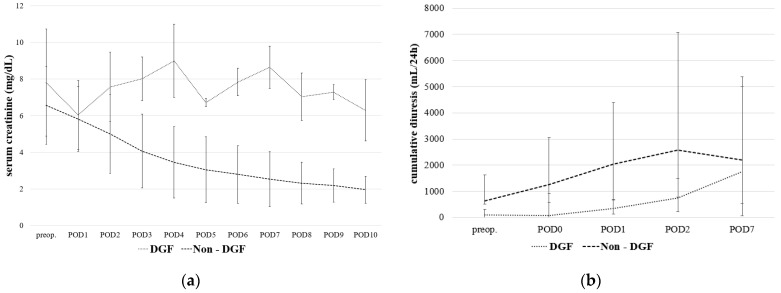
Comparison of average serum creatinine (**a**) and cumulative diuresis (**b**) between patients with normal postoperative graft function (non-DGF) and delayed graft function (DGF) over time (POD, postoperative day) following kidney transplantation. Data are expressed as a mean and standard deviation (**a**) or the median and Q1–Q3 interquartile range (**b**), respectively.

**Figure 3 jpm-12-01749-f003:**
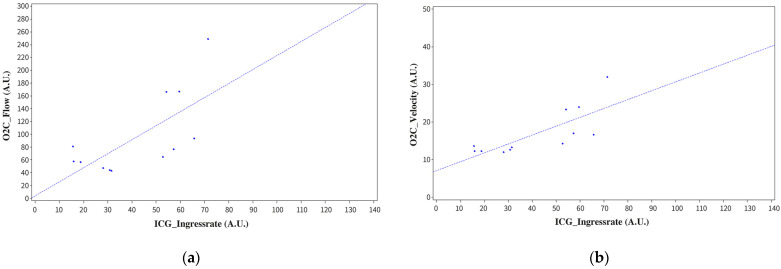
(**a**) Correlation of microvascular flow determined by O2C with ICG IngressRate (r = 0.69319, *p* = 0.0124). (**b**) Correlation of microvascular velocity determined by O2C with ICG IngressRate (r = 0.76678, *p* = 0.0036).

**Figure 4 jpm-12-01749-f004:**
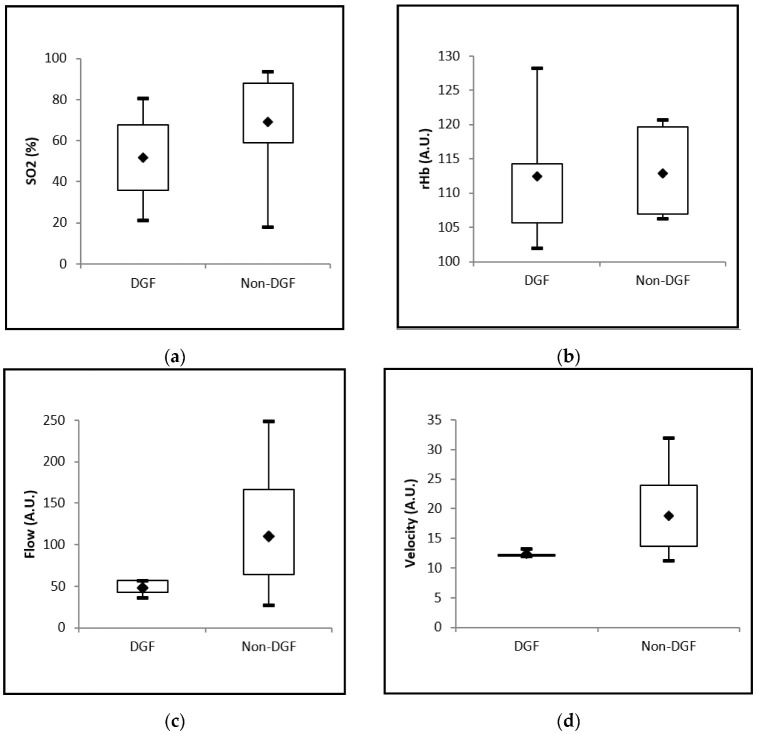
(**a**–**d**) Box plots showing intraoperative microperfusion assessment with O2C in patients with normal as opposed to delayed graft function (DGF) after kidney transplantation with significant differences for the parameters flow and velocity (*p*-values: (**a**), *p* = 0.1700; (**b**), *p* = 0.5323; (**c**), *p* = 0.0314; (**d**), *p* = 0.0171). A.U., Arbitrary Units; flow, microvascular flow; rHb, relative microvascular amount of hemoglobin; SO2, postcapillary oxygen saturation of hemoglobin; velocity, microvascular blood flow velocity.

**Table 1 jpm-12-01749-t001:** Patient and periprocedural characteristics (*n* = 26). Continuous variables are expressed as a mean and standard deviation or as the median with minimum and maximum. For categorical variables, absolute (numbers) and relative (percentage) frequencies are given.

Recipient Characteristics	
Age (years)	56 (±15)
Gender (♀;♂)	6 (23); 20 (77)
Body mass index (kg/m^2^)	25 (±4)
Preoperative eGFR (mL/min/1.73 m^2^)	8.5 (5–26)
Preoperative hemoglobin	12 (±1.8)
Time on dialysis (months)	47 (0–158)
Smoker	4 (15)
Chronic kidney disease stage 4	3 (12)
Chronic kidney disease stage 5	23 (88)
Renal anemia	18 (69)
Diabetes mellitus	7 (27)
Dyslipidemia	12 (48)
Hypertension	23 (88)
Peripheral arterial occlusive disease	1 (4)
Chronic heart failure	3 (12)
Periprocedural characteristics	
Living-donor transplantations	4 (15)
Deceased-donor transplantations (all DBD)	22 (85)
1 artery	20 (77)
2 arteries	6 (23)
Separate pole artery	3 (12)
1 vein	26 (100)
Operating time (minutes)	129 (±47)
Cold ischemia time (minutes)	602 (±308)
Warm ischemia time (minutes)	26 (±8)
Postoperative characteristics	
Diuresis POD 1 (mL/24 h)	1880 (30–9000)
Diuresis POD 2 (mL/24 h)	2240 (100–12,300)
Diuresis POD 7 (mL/24 h)	1770 (1050–5500)

Legend: DBD, donation after brain death; POD, postoperative day.

**Table 2 jpm-12-01749-t002:** Comparison of recipient, donor, and periprocedural characteristics between recipients with normal graft function (Non-DGF, *n* = 21) and those with delayed graft function (DGF, *n* = 5) after kidney transplantation. Continuous variables are expressed as mean and standard deviation. For categorical variables, absolute (numbers) and relative (percentages) frequencies are given.

Recipient Characteristics	DGF	Non-DGF	*p*-Value
Age (years)	61 (±9)	55 (±16)	0.5803
Gender (♀;♂)	0 (0); 5 (100)	6 (29); 15 (71)	0.2981
Body mass index (kg/m^2^)	27 (±2)	25 (±5)	0.3657
Smoker	0	4 (19)	0.5552
Preoperative eGFR (mL/min/1.73 m^2^)	6 (5–24)	9 (5–26)	0.5984
Comorbidities			
Renal anemia	4 (80)	14 (67)	1.0000
Diabetes	2 (40)	5 (24)	0.5875
Dyslipidemia	1 (20)	11 (55)	0.3217
Hypertension	5 (100)	18 (86)	1.0000
Peripheral arterial occlusive disease	0	1 (5)	1.0000
Chronic Heart Failure	0	3 (14)	1.0000
Donor characteristics			
Age (years)	70 (±11)	67 (±14)	0.7641
Gender (♀;♂)	3 (19); 2 (20)	13 (81); 8 (80)	1.0000
First donor creatinine (mg/dL)	1.0 (0.3–1.7)	0.9 (0.4–2.6)	0.5156
Last donor creatinine (mg/dL)	1.0 (0.3–5.2)	0.8 (0.4–3.4)	0.3652
Smoker	1 (20)	3 (30)	1.0000
Cause of death (cerebral hypoxia)	3 (60)	2 (12)	0.0549
Procurement and periprocedural characteristics			
Donation (living; postmortem)	0 (0); 5 (100)	4 (19); 17 (81)	0.5552
Arterial supply 1/2 arteries	4 (80)/1 (20)	16 (77)/5 (24)	1.0000
Operating time (minutes)	130 (95–183)	110 (64–239)	0.3313
Cold ischemia time (minutes)	554 (430–1431)	611 (96–1129)	0.3162
Warm ischemia time (minutes)	21 (16–28)	26 (15–40)	0.5801
Intraoperative urine production (none, moderate or good)	2 (50); 2 (50)	3 (19); 13 (82)	0.2487

**Table 3 jpm-12-01749-t003:** Association between intraoperative perfusion assessment of the allograft and delayed graft function (DGF) after kidney transplantation. Quantitative perfusion assessment was performed with ICG fluorescence angiography and O2C.

Perfusion Parameter	DGF	Non-DGF	*p*-Value
Ingress (A.U.)	172 (100–233)	147.5 (77–252)	0.7500
IngressRate (A.U.)	23.5 (15.9–31.5)	41.8 (8.4–71.5)	0.5819
Egress (A.U.)	54.5 (48–80)	83 (32–196)	0.3963
EgressRate (A.U.)	3.9 (1.3–7.8)	4.7 (2.5–31)	0.2161
SO2 (%)	52.5 (21.3–80.7)	70.3 (18.0–93.7)	0.1700
rHb (A.U.)	111.7 (102.0–128.3)	113.0 (106.3–120.7)	0.5240
Flow (A.U.)	47.0 (36.7–57.3)	81.0 (27.3–249.0)	0.0275
Velocity (A.U.)	12.3 (12.0–13.3)	17.0 (11.3–32.0)	0.0119
Visual * (homogeneous/marbled)	4 (80); 1 (20)	20 (95); 1 (5)	0.3538

* Visual perfusion assessment was performed by the surgeons intraoperatively after reperfusion in accordance with the requirements by the German Organ Procurement Organization (DSO). Continuous variables are expressed as median values together with minimum and maximum. For categorical variables, absolute and relative frequencies are given. Legend: A.U.: arbitrary units; flow, microvascular blood flow; rHb, relative microvascular amount of hemoglobin; SO2, postcapillary oxygen saturation of hemoglobin; velocity, microvascular blood flow velocity.

## Data Availability

The data presented in this study are available from the corresponding author upon request. The data are not publicly available due to ethical restrictions.

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
