# Peer review of "Investigation of Different Methods of Intraoperative Graft Perfusion Assessment during Kidney Transplantation for the Prediction of Delayed Graft Function: A Prospective Pilot Trial"

_jpm, 2022, doi:10.3390/jpm12101749_

Round 1

Reviewer 1 Report

Paper is well written and composed. Identifies aspects that can help to reduce post-kidney allograft dialysis requirements. 

Author Response

Response to Reviewer 1 Comments

Reviewer: Paper is well written and composed. Identifies aspects that can help to reduce post-kidney allograft dialysis requirements. 

Response: Thank you very much for your positive review.

Reviewer 2 Report

The paper "investigation of different methods of intraoperative graft perfusion assessment during kidney transplantation for the prediction of delayed graft function: a prospective pilot trial" highlights a hot topic in organ transplantation such as the assessment of the organs in order to predict the outcome. In this trial, Gerken and colleagues analyze the utilization of ICG and O2C, comparing these techniques to the standard ultrasound and to visual assesment. The paper is well structured and readable.

Minor comments and suggestions:

-line 47: authors should briefly explain why DGF is an increasing clinical problem

-line 131: please modify this sentence (four parameter-> only ingressrate and egressrate are reported and the reader could not easily understand which are the other two parameters)

-line 164: pleace specify the type of donors. Transplant community knows that DCD are not used in Germany. However, it could be better to specify in order to be understandable by everyone.

-line 174 "five"

-If possible, a small paragraph with data about the correlation between ICG and DGF should be added

Author Response

Response to Reviewer 2 Comments

The paper "investigation of different methods of intraoperative graft perfusion assessment during kidney transplantation for the prediction of delayed graft function: a prospective pilot trial" highlights a hot topic in organ transplantation such as the assessment of the organs in order to predict the outcome. In this trial, Gerken and colleagues analyze the utilization of ICG and O2C, comparing these techniques to the standard ultrasound and to visual assesment. The paper is well structured and readable.

Minor comments and suggestions:

Point 1: -line 47: authors should briefly explain why DGF is an increasing clinical problem

Response 1: Thank you very much for this comment. The rising incidence rate of DGF can possibly be explained by the introduction of expanded criteria donors (ECD) and donation after cardiac death (DCD). We have added this sentence in line 48-49, and also added a sentence explaining ECD programs in Europe (ll. 51-53).

Point 2: -line 131: please modify this sentence (four parameter-> only ingressrate and egressrate are reported and the reader could not easily understand which are the other two parameters)

Response 2: We have modified this sencence by introducing quotation marks to highlight the following four parameters: Ingress, Egress, IngressRate, EgressRate.

Point 3: -line 164: pleace specify the type of donors. Transplant community knows that DCD are not used in Germany. However, it could be better to specify in order to be understandable by everyone.

Response 3: In order to clarify this aspect, we have added "from donation after brain death" after "22 deceased donors" in l. 167.

Point 4: -line 174 "five"

Response 4: Thank you very much, we have corrected the spelling mistake accordingly.

Point 5: -If possible, a small paragraph with data about the correlation between ICG and DGF should be added

Response 5: There was no correlation analysis between ICG and DGF, because we evaluated this association by using Fisher's exact test or Mann Whitney U test. Thus, we added the paragraph 3.5 about the association between ICG and DGF (ll. 220-225).